# The Angiopoietin/Tie2 Pathway in Hepatocellular Carcinoma

**DOI:** 10.3390/cells9112382

**Published:** 2020-10-30

**Authors:** Bart Vanderborght, Sander Lefere, Hans Van Vlierberghe, Lindsey Devisscher

**Affiliations:** 1Department of Internal Medicine and Pediatrics, Department of Gastroenterology and Hepatology, Hepatology Research Unit, Ghent University, B-9000 Ghent, Belgium; bart.vanderborght@ugent.be (B.V.); sander.lefere@ugent.be (S.L.); hans.vanvlierberghe@uzgent.be (H.V.V.); 2Department of Basic and Applied Medical Sciences, Gut-Liver Immunopharmacology Unit, Ghent University, B-9000 Ghent, Belgium

**Keywords:** angiopoietin-1, angiopoietin-2, hepatocellular carcinoma, angiogenesis, vascular endothelial growth factor, treatment, biomarker, diagnosis, prognosis

## Abstract

Due to the usually late diagnosis and lack of effective therapies, hepatocellular carcinoma (HCC), which poses a growing global health problem, is characterized by a poor prognosis. Angiogenesis plays an important role in HCC progression, and vascular endothelial growth factor (VEGF) and angiopoietins (Angs) are key drivers of HCC angiogenesis. VEGF-targeting strategies already represent an important component of today’s systemic treatment landscape of HCC, whereas targeting the Ang/Tie2 signaling pathway may harbor future potential in this context due to reported beneficial anticancer effects when targeting this pathway. In addition, a better understanding of the relation between Angs and HCC angiogenesis and progression may reveal their potential as predictive factors for post-treatment disease progression and prognosis. In this review, we give a comprehensive overview of the complex role of Ang/Tie2 signaling in HCC, pinpointing its potential value as biomarker and target for HCC treatments, aiding HCC diagnosis and therapy.

## 1. Hepatocellular Carcinoma

As the sixth-most commonly diagnosed type of cancer and the fastest-rising cause of cancer-related mortality worldwide, hepatocellular carcinoma (HCC), the main primary liver cancer, poses a major global health problem [1]. HCC usually occurs in the setting of chronic liver disease, characterized by chronic hepatic inflammation and fibrosis, and is mainly caused by chronic viral hepatitis, persistent alcohol abuse or metabolic syndrome [2,3]. Due to the worldwide increasing incidences of alcoholic liver disease (ALD) and nonalcoholic fatty liver disease (NAFLD), the burden of HCC is still rising [4,5,6]. HCC is an aggressive cancer with a poor prognosis, as it is often diagnosed at an advanced stage, whereas potential curative interventions, including surgical resection and percutaneous local ablation, are only effective in early stage HCC, and orthotopic liver transplantation is only possible in a limited number of patients [7]. Systemic therapy with multikinase inhibitors, including sorafenib and lenvatinib, only yield minor survival benefits, which are offset by considerable adverse events [4,8]. Even though immune checkpoint inhibitors are promising new drugs in the therapeutic landscape of HCC and have shown a survival benefit when combined with the antivascular endothelial growth factor (VEGF) antibody bevacizumab, there is still an unmet need for more effective treatment strategies, since immunotherapy is not effective or applicable in all HCC patients [9,10]. In addition, the coexistence with chronic liver disease severely hampers the utilization of several biomarkers for the surveillance and early diagnosis of HCC, with alpha-fetoprotein (AFP) to date being the only widely used serum biomarker in daily practice [11]. Thus, it is of the utmost importance to continue refining and broadening the current understanding of the complex pathogenesis of HCC to enable the discovery of new therapeutic targets and biomarkers for early HCC diagnosis.

## 2. Angiogenic Sprouting in HCC

Angiogenesis plays a major role in the growth, progression and metastasis of HCC, a characteristically hypervascular tumor [12,13]. Indeed, in order to meet the increased oxygen and nutrient consumption of this fast-growing tumor, the formation of new blood vessels is essential [12,14]. In neovascularization, endothelial cells of pre-existing vessels proliferate and migrate to form vascular sprouts [15]. During HCC growth and dedifferentiation, a switch from portal venous to predominantly arterial blood supply occurs, and well-developed artery-like vessels characterize angiogenic sprouting [12,16]. This complex multi-step process is induced by a balanced shift between multiple pro- and antiangiogenic factors, known as the angiogenic switch, with VEGF and angiopoietins (Angs) as key contributors to HCC angiogenesis [15,16,17,18]. Both of these angiogenic growth factor families serve as ligands of receptor tyrosine kinases, which are almost exclusively expressed on the surface of endothelial cells, and they have complementary and coordinated functions in neovascularization [16,19,20]. VEGF mediates angiogenesis by inducting endothelial cell proliferation, migration and tube formation, as well as by increasing the vascular permeability [16,21]. The expression of VEGF and its receptors is upregulated in various human cancers, including HCC, and often correlates with microvessel density (MVD), invasiveness and poor prognosis [21,22]. Consequently, anti-VEGF strategies have been extensively explored and represent an important component of today’s systemic anticancer therapy [23]. In this regard, a combination therapy of the VEGF-targeting monoclonal antibody bevacizumab with the immune checkpoint inhibitor (ICI) atezolizumab has shown very promising effects on the survival of unresectable HCC patients in a phase 3 trial [10]. This synergistic antitumor effect of combined VEGF target and ICI therapy may be explained by the vasculature normalization established by the antiangiogenic agent, which improves the efficacy of the immunotherapeutic compound [10,24,25]. The other key factor in HCC angiogenesis, Ang-2, also exerts immunosuppressive activities via involvement in the recruitment of monocytes/macrophages in the tumor microenvironment (TME) and induction of the expression of the immune checkpoint programmed death-ligand 1 (PD-L1) on the surface of tumor-associated macrophages (TAMs) and may thus also contribute to the resistance to ICI therapy. As Ang-2 mediates resistance to anti-VEGF therapy as well, and Ang-2 targeting may promote restoration of the hepatic vasculature, the addition of an Ang-2-targeting compound to the combination of anti-VEGF and ICI therapy could potentially further improve its efficacy in HCC patients [26]. The Ang/tyrosine kinase with immunoglobulin (Ig) and epidermal growth factor (EGF) homology domains 2 (Tie2) system thus exerts a critical role in neovascularization in conjunction with VEGF and might also present an interesting therapeutic target for HCC, as discussed further (Figure 1) [16,27].

## 3. Role of Angiopoietins in Angiogenic Sprouting

Over two decades ago, Yancopoulous et al. discovered a novel family of growth factors specifically binding to an endothelial cell-expressed receptor tyrosine kinase, consisting of four members: Angs 1–4 [29,34,35,36]. Despite all of these being ligands of the Tie2 receptor, a ligand-receptor interaction leads to different subtype-specific biological actions [37]. The exact functions of the murine Ang-3 and its human ortholog Ang-4 are poorly characterized, while Ang-1 and Ang-2 are considered the two pivotal angiogenesis-mediating members of the Ang family [15,37]. Ang-1 is widely expressed in tissues during adult life, activates the Tie2 receptor via phosphorylation and mediates the stabilization and maturation of developing vessels by strengthening endothelial cell-cell junctions and by promoting the recruitment of pericytes and smooth muscle cells [20,22]. In contrast, Ang-2 is primarily expressed in endothelial cells, where it is stored in specialized Weibel-Palade bodies from which it can be rapidly released upon cytokine stimulation. Ang-2 has a similar Tie2 receptor affinity as Ang-1 and is a context-dependent Tie2 agonist/antagonist that activates the Tie2 receptor under homeostatic conditions and inhibits Tie2 phosphorylation in the presence of inflammation [15,38]. However, both Ang-2-mediated overactivation and inactivation of Tie2 can contribute to vascular remodeling [39]. Under physiologic conditions, Ang-2 expression is markedly increased only at sites of active vascular remodeling, such as the female reproductive tract [16,20]. On the other hand, Ang-2 is overexpressed in a wide range of inflammatory conditions, including HCC, which is a typical inflammation-associated cancer [40,41]. Ang-2/Tie2 signaling results in vessel destabilization, which facilitates proangiogenic factors, including VEGF, to induce neovascularization [32,33]. Indeed, in the presence of VEGF, Ang-2 mediates vascular sprouting and angiogenesis [22]. However, as Angs themselves do not promote endothelial cell proliferation, Ang-2/Tie2 signaling results in vessel regression in the absence of VEGF [15,16,27,33]. In a variety of human malignancies, including HCC, Ang-2 overexpression is correlated with MVD, several clinicopathological parameters and poor prognosis, as further discussed [15,22,32].

## 4. Angiopoietin Expression and Signaling in HCC

### 4.1. Angiopoietin-2 Expression in HCC

In 1999, Tanaka et al. were the first to show a close association between Ang-2 expression and hypervascularity in human HCC. In addition, they demonstrated that Ang-2 overexpression promotes rapid tumor development and worsens prognosis in an ectopic xenograft model of human HCC [32]. Correlation of the overexpression of both Ang-2 and Tie2 with angiogenesis, as well as certain histopathological parameters in human HCC, confirmed that Ang-2/Tie2 angiogenic signaling may be implicated in the biological behavior of HCC [42]. Moreover, inhibition of the Tie2 receptor suppressed tumor neovascularization and growth in an HCC mouse model, leading to the suggestion that, in addition to serving as a Tie2 antagonist, Ang-2 may possibly also activate Tie2 as an agonist, resulting in stimulation of a fundamentally different signaling pathway compared to Ang-1 [19]. Later on, Yoshiji et al. demonstrated the VEGF dependency of Ang-2 to exert its effect on HCC angiogenesis and development. Whereas a combined overexpression of Ang-2 and VEGF in HCC resulted in markedly increased tumor development and neovascularization, significant upregulation of matrix metalloproteinase (MMP)-2 and MMP-9 expression and a marked reduction of intratumoral apoptosis and vessel maturation; these effects were not observed or far less pronounced following the overexpression of, respectively, Ang-2 and VEGF alone. Moreover, the inhibition of VEGF signaling abrogated the effect of combined Ang-2 and VEGF overexpression on HCC tumor development. These findings lead to the suggestion that the synergistic effect of Ang-2 and VEGF on HCC angiogenesis and development is partly mediated by reduced intratumoral apoptosis and the failure of tumor vessel maturation, as well as an induction of MMP-2 and MMP-9 [27].

Several studies have further investigated the underlying regulatory pathways of Ang-2. As hypoxia is considered as an important stimulus of angiogenic signaling in cancer and is involved in the upregulation of VEGF in HCC; Ang-2/Tie2 signaling might also be hypoxia-driven in HCC [17]. Several groups did not observe differences in either Ang-1 or Ang-2 mRNA expression in human HCC cell lines under hypoxic conditions and suggested that hypoxia does not regulate Ang expression in HCC [17,33]. However, Ang-2 release from Weibel-Palade bodies is increased in hypoxia-stimulated endothelial cells, and this might contribute to Ang-2 overexpression in the hypoxic microenvironment of HCC tissue [28]. Furthermore, VEGF has been shown to upregulate the expression of Ang-1 and Ang-2; thus, hypoxia-induced VEGF overexpression may also be involved in Ang-2 overexpression in HCC [29].

Tanaka et al. suggested a regulatory role of cyclooxygenase (COX)-2 in the expression of Ang-2 in HCC, since specific COX-2 inhibition attenuated Ang-2 expression, as well as inhibited tumor angiogenesis and growth in a subcutaneous syngeneic HCC mouse model. Therefore, although further exploration of the underlying regulatory mechanisms is needed, modulation of Ang-2 regulators, including COX-2, might be a potential therapeutic strategy in HCC [30].

Recent advances in cancer exome sequencing have led to the identification of AT-rich interactive domain-containing protein 1A (Arid1a), a key member of the switch/sucrose nonfermentable (SWI/SNF) chromatin-remodeling complex, as one of the genes that is most frequently mutated in human HCC, present in approximately 10–15% of the patients [31,43]. Arid1a deficiency resulted in the upregulation of Ang-2 in HCC tissue through histone H3K27ac modification at the Ang-2 gene locus. Importantly, sorafenib is able to counteract this protumorigenic effect in Arid1a-deficient HCC by reducing H3K27ac deposition at the Ang-2 gene locus and, consequently, epigenetically downregulating Ang-2 expression. As Arid1a deficiency seems to result in higher susceptibility to sorafenib treatment, screening for this mutation may enable the specific selection of HCC patients who are more likely to benefit from this systemic therapy [31].

### 4.2. Angiopoietin-1 Expression in HCC

Several groups demonstrated similar Ang-1 expression in HCC and adjacent nonmalignant hepatic tissue [21,32]. As Ang-1 and Ang-2 compete for interaction with the same receptor, and Ang-2 expression is upregulated in human HCC tissue, Mitsuhashi et al. suggested that the quantitative balance between Ang-1 and Ang-2 expression might be a better way to assess the role of Ang signaling on HCC angiogenesis and progression. They demonstrated that, in the presence of VEGF, the Ang-2/Ang-1 mRNA ratio was indeed closely associated with angiogenesis, several clinicopathological parameters and a poor prognosis in HCC [21]. Several groups did observe an overexpression of Ang-1 in human HCC samples. However, in contrast to Ang-2 expression, Ang-1 upregulation did not correlate with angiogenesis or tumor progression, strengthening the proposal of Mitsuhashi et al. to consider the Ang-2/Ang-1 ratio instead of focusing on individual expressions of the two Ang subtypes [20,33]. Sugimachi et al. demonstrated that the progression from normal hepatic tissue to HCC is accompanied by an induced expression of Ang-2 and suppression of Ang-1 expression. However, progression to a less-differentiated tumor seems to be associated with an upregulation of both Ang subtypes. Therefore, as the Ang-2 expression is higher at the invasive front of the HCC tissue, compared to the center of the tumor, and Ang-2 overexpression contributes to VEGF-mediated neovascularization of the growing tumor; the upregulation of Ang-1 might promote stabilization and maturation of the newly-formed vessels [17].

Lin et al. also observed Ang-1 overexpression in HCC tissue. As previous studies had reported that HCC progression is promoted by activated hepatic stellate cell (HSC)-mediated microvessel formation, and the expression of Ang-1 was positively correlated with the MVD and α-smooth muscle actin (α-SMA) expression, the authors suggested that activated HSCs mediate angiogenesis through the upregulation of Ang-1 expression and, thereby, promote HCC growth and metastasis [44]. An obvious shortcoming of this study is the fact that the authors did not determine the intratumoral Ang-2 expression and, thus, had no insight into the quantitative balance between both Ang subtypes, and the resultant nature and magnitude of the Ang/Tie2 signal.

## 5. Role of Angiopoetin-2 in Nonsprouting Vascular Remodeling in HCC

In stark contrast to multiple studies cited above, Zeng et al. did not detect convincing changes in either Ang-1 or Ang-2 expression in HCC tissue and observed an Ang-2/Ang-1 mRNA ratio that was multiple times lower compared to renal cell carcinoma (RCC), one of the most highly angiogenic human cancers. Therefore, they proposed that Ang/Tie2-mediated angiogenesis may not play a major role in HCC tumorigenesis. Moreover, in their study, VEGF expression was not upregulated as well, strengthening their hypothesis that HCC-mediated vascular remodeling may rely on other mechanisms than angiogenic sprouting [45]. Two such additional mechanisms for tumor blood supply that have been described in HCC are sinusoidal capillarization and vessel co-option. Sinusoidal capillarization, which has been found to be a common neoangiogenic process in HCC, implies a transformation of the discontinuous hepatic sinusoids into continuous capillaries. Well-differentiated HCC tissue contains microvessels of both the sinusoidal and capillary type, whereas only the capillary type is present in poorly differentiated HCC, so sinusoidal capillarization appears to be associated with tumor dedifferentiation [12]. Ang-1 may be involved in the stabilization and maturation of these vessels, by promoting pericyte recruitment [45]. In vessel co-option, tumor cells grow along the pre-existing vasculature without eliciting an angiogenic response. However, as the tumor grows, the co-opted vessels will upregulate Ang-2 expression, which, due to the lack of VEGF expression, first leads to vascular regression. The subsequent formation of a hypoxic core stimulates coincident VEGF upregulation and induces angiogenesis [46,47]. Thus, alternative mechanisms for vascular remodeling may also be implicated in HCC tumorigenesis. The involvement of each of these mechanisms may shift during HCC development and progression, depending on the balance of proangiogenic and antiangiogenic factors.

Fang et al. described the existence of a vascular pattern of sinusoid-like vessels that closely surrounds individual tumor clusters in HCC, named vessels that encapsulate the tumor cluster (VETC). They found that the development of this vascular pattern depends on the presence of intratumoral Ang-2, as VETC-positive HCC tissue displays a higher Ang-2 expression and the intratumoral inhibition of Ang-2 disrupts the formation of this VETC pattern in vivo. Moreover, Ang-2 inhibition also suppresses the in vivo metastasis of HCC, suggesting that this Ang-2-dependent VETC pattern is involved in HCC metastasis [48]. Subsequently, Zhou et al. identified two microRNAs (miRNAs) that act as suppressors of VETC formation and VETC-dependent metastasis, through the inhibition of Ang-2 expression, either directly in case of miR-125b or, indirectly, through suppression of the mammalian target of the rapamycin (mTOR)-p70S6K signaling pathway in case of miR-100. HCC patients with low levels of these miRNAs have a high intratumoral Ang-2 expression and may therefore be more susceptible to tumor metastasis. Consequently, Ang-2 targeting may disrupt the formation of VETC, and miR-125b and miR-100 may represent new promising targets for the prevention of HCC metastasis [49].

## 6. Angiopoietin-Targeting Therapeutic Strategies in HCC

### 6.1. Angiopoietin Targeting in Context of Chemotherapy

Several chemotherapeutic strategies have been shown to influence Ang-2/Tie2 signaling in HCC. Wada et al. demonstrated the therapeutic potential of combined interferon gamma (IFN-γ) and fluorouracil (5-FU) treatments by inhibiting tumor cell proliferation and angiogenesis and inducing intratumoral apoptosis. These effects were accompanied by a significant decrease in Ang-2 expression and increase in Ang-1 expression in an HCC mouse model [50]. The authors suggested that the enhanced antitumorigenic effect of 5-FU may be partially explained by an improvement of its delivery to the HCC tumor, as IFN-γ-mediated Ang regulation may cause intratumoral vessel stabilization and the reduction of vascular permeability [51]. Similarly, a significant downregulation of Ang-2 expression was also observed following interstitial chemotherapy using poly(lactic-co-glycolic acid) (PLGA) microspheres containing docetaxel in an HCC mouse model, suggesting that inhibition of Ang-2/Tie2-mediated angiogenesis may be an important mechanism involved in its antitumor activity [52].

Li et al. were the first to demonstrate that Ang-2 is able to enhance chemoresistance in HCC. In a human HCC cell line, they showed that Ang-2/Tie2 signaling attenuates doxorubicin-induced apoptosis. This antiapoptotic effect is mainly achieved through prevention of doxorubicin-induced oxidative stress and mitochondrial dysfunction. The underlying mechanism of this Ang-2-mediated chemoresistance is upregulation of the expression of survivin and Ref-1 through the Tie2-extracellular signal-regulated kinase (ERK)-mitogen- and stress-activated protein kinase (MSK) cascade. Ang-2 targeting thus not only represents a promising antiangiogenic therapeutic strategy but may, in addition, also be of interest to increase chemosensitivity in HCC patients [53]. The additional finding that a Tie2-neutralizing antibody led to reversal of the Ang-2-mediated chemoresistance suggests that, in addition to serving as a biological antagonist of Ang-1, Ang-2 can also act as a context-dependent Tie2 agonist, which is something that was already previously put forward by Tanaka et al. [19,53].

### 6.2. Direct Angiopoietin-Targeting Therapeutic Strategies

Agents that directly target Ang-2 expression are emerging. In 2014, Zhang et al. were the first to demonstrate the effect of direct Ang-2 targeting on HCC angiogenesis and progression in vivo. For this, they purified a single-chain variable fragment against human Ang-2 (scFv-Ang2) that significantly reduced endothelial migration and tubule formation in vitro and investigated its antitumor potential in a highly metastatic murine orthotopic xenograft model of human HCC. scFv-Ang2 treatment led to significant inhibition of HCC angiogenesis and growth, as well as partial inhibition of intrahepatic metastasis and metastatic spread to the lungs [54]. These beneficial effects on in vivo HCC angiogenesis and progression suggest that the direct targeting of Ang-2 could be an effective therapeutic strategy in HCC patients. T7 peptide, a fragment of the endogenic antiangiogenic factor tumstatin, has also been investigated as an antimetastatic inhibitor of Ang-2 in the context of HCC. The administration of T7 peptide downregulated the hypoxia-induced expression of Ang-2 in endothelial cells through the inhibition of Akt (protein kinase B) phosphorylation. This resulted in the inhibition of both angiogenesis and HCC tumor cell invasion. However, as these promising antitumor properties of T7 peptide were only demonstrated in vitro, they must be validated in a subsequent in vivo study in order to prove its therapeutic potential in HCC [28]. Recently, a study from our group showed that Ang-2 inhibition prevents the progression of chronic liver disease towards HCC. In addition to its beneficial effect on steatohepatitis and fibrosis, the treatment with the Ang-2/Tie2 interaction-inhibiting peptibody L1-10 in a mouse model of diabetes-associated NAFLD attenuated HCC development [38]. In order to further explore the role of Ang-2 in the progression of chronic liver disease, it would be interesting to investigate the effects of Ang-2 inhibition on HCC development in experimental models of ALD and chronic liver disease due to hepatitis B virus (HBV) or hepatitis C virus (HCV) infections. Table 1 summarizes the experimental HCC animal models used in Ang-related research to date. Most of these HCC models are implantation models in which murine or human HCC cells are orthotopically or subcutaneously injected in mice.

The human anti-Ang-2 monoclonal antibody nesvacumab was the first direct Ang-2-targeting therapy in a first-in-human phase I study (NCT01271972) in patients with advanced solid tumors. In this clinical trial, two HCC patients showed stable disease, with tumor regression and significant AFP decline [55].

**Table 1 cells-09-02382-t001:** Hepatocellular carcinoma (HCC) animal models used in angiopoietin-related research to date.

Experimental HCC Animal Model	Ref.
Diethylnitrosamine-induced HCC rat model	[56]
Diethylnitrosamine-induced HCC mouse model	[31]
Nonalcoholic steatohepatitis-induced HCC mouse model	[38]
Subcutaneous syngeneic HCC mouse model (injection of MH134 cells in C3H mice)	[30]
Subcutaneous syngeneic HCC mouse model (injection of Hepa1-6 cells in C57BL/6 mice)	[48,49]
Orthotopic syngeneic HCC mouse model (injection of Hepa1-6 cells in C57BL/6 J mice)	[48,49]
Murine subcutaneous xenograft model of human HCC (injection of BEL-7404 cells in BALB/c nude mice)	[52]
Murine orthotopic xenograft model of human HCC (injection of VETC-2 cells in BALB/c nude mice)	[49]
Highly metastatic murine orthotopic xenograft model of human HCC (implantation of metastatic tumor tissue of murine subcutaneous xenograft model of human HCC in BALB/c nude mice)	[54]

## 7. Angiopoietins as Diagnostic and Prognostic Biomarkers

Scholz et al. were the first to show that, in addition to an upregulation in HCC tissue, Ang-2 levels are also elevated in the blood of HCC patients, leading to its potential as a serum biomarker for HCC. However, no correlation between Ang-2 serum levels and tumor characteristics could be demonstrated. As they already observed elevated Ang-2 serum levels in cirrhotic patients, the authors speculated that locally and systemically increased Ang-2 levels may contribute to the development of HCC in patients suffering from cirrhotic liver disease [57]. Kuboki et al. showed a significant correlation between hepatic (but not peripheral) venous Ang-2 levels and Ang-2 mRNA expression and the MVD in HCC tissues, suggesting that hepatic venous Ang-2 levels reflect HCC neovascularization more precisely compared to peripheral venous levels. However, also for hepatic venous Ang-2 levels, no correlation with clinicopathological parameters of HCC was found, except for portal vein invasion [58]. In contrast, other groups observed strong correlations between systemic Ang-2 levels and tumor stage and the Child-Pugh score [13,59]. Llovet et al. reported on Ang-2 plasma levels as a strong, independent prognosis predictor in patients with HCC, since high Ang-2 plasma levels correlated with poor survival, shorter time to progression and several other clinical/demographic variables associated with a poor outcome in advanced HCC [60].

In 2016, a five-gene transcriptomic hepatic signature, which includes Ang-2 as the most significantly upregulated gene, in addition to delta-like ligand 4 (DLL4), neuropilin/tolloid-like 2 (NETO2), endothelial cell-specific molecule-1 (ESM1) and nuclear receptor subfamily 4 group A member 1 (NR4A1), was identified to predict HCC growth and prognosis in individual patients. Adding the prognostic information of this five-gene signature to the parameters that are already used in daily practice could lead to a more personalized therapeutic management of HCC patients by allowing better stratification of these patients into clinically relevant subgroups [61].

Teixeira et al. recently demonstrated the differential expression of Ang-2 in HCC lesions compared to non-neoplastic regenerative nodules. They suggested that, in addition to its clear potential as a predictive or prognostic biomarker, Ang-2 might also serve as a diagnostic biomarker in the pathologic interpretation of malignancy in hepatocellular nodules. Anti-Ang-2 immunohistochemistry (IHC) might, for instance, be added to the current triple-IHC panel for HCC diagnosis, consisting of heat shock protein 70 (Hsp70), glypican-3 and glutamine-synthetase, and thereby contribute to the handling of challenging cases [62].

### 7.1. Angiopoietins as Biomarker in Nonsystemic HCC Treatment

If HCC is diagnosed at an early stage in patients with a reasonable liver function, the preferred treatment is surgical resection [7]. Whereas several groups showed that Ang-2 expression in patient HCC tissues is predictive for recurrence after surgical resection [22,63], Kuboki et al. were the first to demonstrate that high preoperative hepatic venous Ang-2 levels correlate with a shorter postoperative survival, strengthening the prognostic value of Ang-2 [58]. Diaz-Sanchez et al. confirmed this correlation between high hepatic venous Ang-2 levels and the unfeasibility of both surgical and locoregional curative treatments [13]. A high preoperative Ang-2 level in the hepatic vein may thus reflect a more advanced stage of HCC, in which surgical resection and other curative treatments are ineffective [7,22,58]. Following resection, Ang-2 levels in both the hepatic and peripheral vein decreased significantly [58].

Orthotopic liver transplantation represents the best curative option for early stage HCC patients with underlying cirrhosis [7]. Atanasov et al. assessed the influence of Ang expression and the presence of Tie2-expressing monocytes (TEMs) in the explanted liver prior to liver transplantation on graft rejection and patient survival. Pretransplant Ang-2 expression was associated with graft rejection after transplantation. Moreover, the presence of TEMs was associated with lower post-transplant survival [64]. Therefore, Ang-2 and TEMs could serve as predictive serum biomarkers for the stratification of patients who are more likely to benefit from liver transplantation [65,66].

Once HCC has progressed to an intermediate stage, transcatheter arterial chemoembolization (TACE) represents the therapeutic gold standard for patients without concurrent portal vein thrombosis [67]. This locoregional therapy is based on specific occlusion of the arterial blood supply of the tumor, which progressively develops during HCC dedifferentiation, combined with injection of chemotherapeutic drugs [7,67]. Contradictory findings have been reported on the effect of TACE on circulating Ang-2 levels. Hsieh et al. observed a significant TACE-induced increase in Ang-2 serum levels, whereas Diaz-Sanchez et al. did not notice statistically different levels of circulating Ang-2 after locoregional HCC treatment with radiofrequency ablation (RFA) or TACE [13,59].

As an alternative for TACE, transarterial radioembolization (TARE) can be considered for intermediate-stage HCC [7]. In this locoregional therapy, small radioisotope-containing microspheres are delivered directly into the tumor through its arterial blood supply [7,68]. A major advantage over TACE is the fact that portal vein thrombosis is not a contraindication for TARE [7]. Carpizo et al. detected higher baseline Ang-2 serum levels in liver cancer patients with shorter survival after yttrium-90 (90Y) radioembolization therapy, making it a potentially useful biomarker for post-treatment prognosis. Increased Ang-2 expression might thus indicate a worse survival of HCC patients undergoing TARE by either promoting neovascularization or by negatively influencing the effectiveness of TARE [68].

### 7.2. Angiopoietins as a Biomarker in Sorafenib Treatment

The oral multikinase inhibitor sorafenib was the first systemic therapeutic option to gain approval for use in HCC patients [4]. Despite the minor survival benefit, and considerable adverse events to date, it is still considered as the first-line treatment for unresectable advanced HCC, together with lenvatinib [4,8,69]. Sorafenib exerts its antitumor effect through the inhibition of several targets involved in HCC angiogenesis and progression, including vascular endothelial growth factor receptors (VEGFRs) 1-3, platelet-derived growth factor receptor beta (PDGFR-β), mast/stem cell growth factor receptor (c-KIT) and rearranged during transfection (RET), as well as several rapidly accelerated fibrosarcoma (Raf) kinases [4,7,69]. As a portion of HCC patients does not show a response to this antiangiogenic treatment, the identification of angiogenesis-related serum biomarkers to predict which patients are likely to benefit from this therapy is an area of active interest [69,70]. Miyahara et al. were the first to demonstrate a negative correlation between circulating Ang-2 levels and drug response and post-treatment progression-free survival in sorafenib-treated HCC patients. Measurement of baseline Ang-2 serum levels predicted the efficacy of sorafenib treatment in advanced HCC patients [70]. In a large biomarker study, Llovet et al. showed that sorafenib treatment is able to halt the disease progression-associated increase in circulating Ang-2 in patients with advanced HCC. As increases in circulating Ang-2 during sorafenib treatment were associated with a poor outcome, measuring the serum level of this angiogenic factor may be of interest in disease monitoring during treatment [60]. In addition to the potential of circulating Ang-2 as a prognostic biomarker in sorafenib treatment of advanced HCC patients, recently, three ANGPT2 (Ang-2 gene) rs55633437 single-nucleotide polymorphisms (SNPs) have been identified that correlate with patient survival and response to sorafenib. These ANGPT2 polymorphisms may thus potentially serve as genetic biomarkers for HCC patient stratification [71].

### 7.3. Angiopoietins as Biomarkers in Other Systemic Treatments

As high-circulating Ang-2 levels are associated with poor prognosis, addition of the Ang-1 and Ang-2-neutralizing peptibody trebananib to the sorafenib treatment could improve the survival of advanced HCC patients. However, this combination of antiangiogenic agents did not demonstrate improved disease control. A possible explanation for this lack of additional disease improvement is the potential existence of a nonexceedable maximum benefit from antiangiogenic therapy. If this is the case, in order to achieve further progress in advanced HCC treatment, antiangiogenic agents should be combined with therapeutic strategies that target other cancer hallmarks, such as immunotherapeutic approaches [72]. In this regard, a combination therapy of the immune checkpoint inhibitor atezolizumab and the VEGF-targeting monoclonal antibody bevacizumab in patients with unresectable HCC resulted in superior survival outcomes compared to sorafenib in a phase 3 trial and will, therefore, become the new standard of care in this setting [10,73]. In the case of failure of sorafenib treatment, regorafenib is a second-line systemic treatment option [4]. Interestingly, in addition to sharing common targets with sorafenib, such as VEGFRs 1-3, PDGFR-β and RET, regorafenib also inhibits Tie2. Recently, Teufel et al. identified low baseline circulating Ang-1 levels as predictive for higher overall survival after regorafenib treatment [74].

In addition to these multikinase inhibitors that are already approved for first- or second-line treatment of advanced HCC, several other antiangiogenic therapies have been or are being tested in clinical trials [75]. Concerning the utilization of circulating Ang-2 levels as a predictive or prognostic biomarker in advanced HCC, Kaseb et al. reported that elevated Ang-2 plasma levels were associated with poor prognosis in advanced HCC patients treated with a combination of the bevacizumab and the oral tyrosine kinase inhibitor erlotinib [76]. Kang et al. also observed a correlation between high baseline circulating Ang-2 levels and poor prognosis after the second-line treatment of advanced HCC patients with the tyrosine kinase inhibitor axitinib [77].

In Table 2, we provide an overview of existing indications for Angs to serve as a biomarker in HCC diagnosis or treatment.

## 8. Alternative Angiopoietin-2-Related Targets and Biomarkers in HCC

Tie2-expressing monocytes (TEMs) are a proangiogenic subpopulation of peripheral and tumor-infiltrating myeloid cells, which have been reported in several angiogenesis-mediated cancers, including renal, colorectal, pancreatic and lung carcinoma, as well as HCC. The number of TEMs, characterized as CD14+CD16+Tie2+ monocytes, is significantly increased in blood and tumoral tissues of HCC patients and is positively correlated with Ang-2 expression and angiogenesis in HCC tissues. In the liver, TEMs preferentially accumulate in the perivascular areas of HCC tissues. These findings suggest that TEMs may contribute to HCC-mediated angiogenesis [65,82]. As Matsubara et al. demonstrated, a positive correlation between the frequency of TEMs in the blood and the severity of liver disease and post-therapy recurrence rate, TEM frequency could represent a prognostic biomarker in HCC patients [65]. He et al. also suggested that circulating TEMs may predict the prognosis of patients with HCC, as they observed a negative correlation between TEM percentages in the blood and post-resection survival [82]. TEM frequency also changes dynamically in relation to curative surgical or locoregional treatment and post-treatment recurrence, adding to its prognostic value [65]. Furthermore, the predictive power of circulating TEMs was confirmed in liver transplantation and sorafenib-treated HCC patients as well, as, in both cases, TEMs are associated with survival [64,83]. In addition, the frequency of circulating TEMs was found to be superior to levels of the HCC-specific serum marker AFP in differentiating HCC from chronic liver disease or cirrhosis and may, thus, also represent a complementary diagnostic biomarker for HCC [65].

Very recently, Xie et al. described the existence of Ang-2-expressing HCC-derived exosomes. These exosomes are internalized and recycled in endothelial cells, and thus, exosomal Ang-2 interacts differently with its recipient cells compared to soluble Ang-2. HCC cell-secreted exosomal Ang-2 seems to promote HCC angiogenesis and progression independent of the Ang-2/Tie2-signaling pathway by activation of the Akt/endothelial nitric oxide synthase (eNOS) and Akt/β-catenin pathways. Consequently, the existence of these alternative signaling pathways for exosomal Ang-2 reveals novel potential targets for the treatment of HCC [79].

## 9. Conclusions

The role of Ang-2/Tie2 signaling in HCC angiogenesis and progression is complex. In addition to its proportional expression compared to Ang-1, Ang-2 is dependent on the presence of VEGF, and the overall balance between proangiogenic and antiangiogenic factors, to contribute to HCC angiogenesis. Ang-2 expression is regulated by multiple mechanisms, and Angs exert several functions through alternative Ang/Tie2-independent signaling pathways. Multiple therapeutic approaches have been demonstrated to target Ang-2, either directly or indirectly, in HCC. However, to date, none of these approaches has resulted in a truly satisfying antiangiogenic treatment for advanced HCC, again illustrating the complexity of its angiogenic landscape. The potential of Ang-2 as a prognosis predictor is shown for a broad range of therapeutic strategies and HCC stages, ranging from surgical approaches in early HCC to systemic treatment in advanced HCC. The potential of Ang-2 as a diagnostic biomarker is, despite its association with poor prognosis, however, less evident, since Ang-2 expression is already upregulated in patients with chronic liver disease.

## Figures and Tables

**Figure 1 cells-09-02382-f001:**
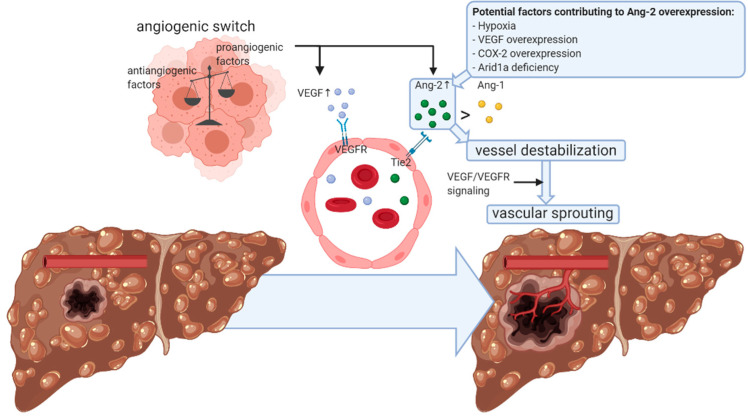
Angiopoietin-2-mediated vascular sprouting in hepatocellular carcinoma. Vascular remodeling is crucial for the growth and progression of hepatocellular carcinoma (HCC) [12,13]. The formation of new blood vessels is induced by the angiogenic switch, which implies an intratumoral balance shift in favor of proangiogenic factors, including vascular endothelial growth factor (VEGF) and angiopoietin (Ang)-2 [15,16,17,18]. Overexpression of Ang-2 may be mediated by hypoxia [28], VEGF overexpression [29], cyclooxygenase (COX)-2 overexpression [30] and AT-rich interactive domain-containing protein 1A (Arid1a) deficiency [31], among others, and alters the quantitative balance between Ang-1 and Ang-2 expression in favor of Ang-2 [21]. Subsequent interaction with its receptor, tyrosine kinase with immunoglobulin (Ig) and epidermal growth factor (EGF) homology domains 2 (Tie2), which are predominantly expressed on the surface of endothelial cells, result in vessel destabilization and, thereby, facilitates other proangiogenic factors, including VEGF, to induce vascular sprouting [32,33]. Ang-2 is thus dependent on VEGF/VEGF receptor (VEGFR) signaling to exert its effect on HCC angiogenesis and progression [27]. Despite the complexity of Ang-2-mediated HCC neovascularization, Ang-2 could potentially serve as a circulating or tissue biomarker for HCC, and therapeutic opportunities lie in direct or indirect targeting of this Ang-2/Tie2 signaling pathway. This figure was created with BioRender.com.

**Table 2 cells-09-02382-t002:** Biomarker potential of tissue and circulating angiopoietins in HCC diagnosis and treatment. Ang: angiopoietin.

Diagnosis/Treatment	Biomarker Potential	Ref.
Diagnosis	Ang-2 expression is higher in HCC tissue compared to adjacent noncancerous liver tissue.	[15,20,21,32,33,42,63]
	Circulating Ang-2 levels are higher in HCC patients compared to cirrhosis patients.	[57,78]
	Ang-2 expression is higher in HCC tissue compared to benign liver disease tissue.	[79]
	Differential Ang-2 expression in HCC lesions, compared to non-neoplastic regenerative nodules.	[62]
Surgical resection	Ang-2 expression in HCC tissue correlates with post-surgery recurrence.	[22,63]
	Preoperative hepatic venous Ang-2 levels inversely correlate with post-surgery survival.	[58]
Liver transplantation	Ang-2 expression in HCC tissue correlates with post-surgery graft rejection.	[64]
Transarterial radioembolization	Baseline circulating Ang-2 levels inversely correlate with post-treatment survival.	[68]
Surgical or locoregional treatment	Circulating Ang-2 levels inversely correlate with eligibility for surgical or locoregional treatment.	[13]
Sorafenib	Baseline circulating Ang-2 levels inversely correlate with response to sorafenib.	[70]
	Baseline circulating Ang-2 levels inversely correlate with post-treatment time to progression and overall survival.	[60,70,80,81]
	Post-treatment increases of circulating Ang-2 levels inversely correlate with time to progression and overall survival.	[60]
Regorafenib	Baseline circulating Ang-1 levels inversely correlate with post-treatment overall survival.	[74]

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
