# Peer review of "The Angiopoietin/Tie2 Pathway in Hepatocellular Carcinoma"

_cells, 2020, doi:10.3390/cells9112382_

Round 1
Reviewer 1 Report
This review paper discussed about the angiopoietin/Tie2 pathway in HCC. Generally it is well-written.
Comments
- An illustrative figure portraits the complexity of this pathway and its associated pharmacological intervention (druggable node) will be helpful for readers.
- Summary subtitles can be placed ahead of each discussion main point to facilitate comprehension.
- The rationale in line 54 and 55 may be misleading. A lot of clinical data showed that a single large HCC mass can be a well-differentiated tumor, instead of showing features of undifferentiation or dedifferentiation.
- Can you discuss, with the molecular insights, why bevacizumab alone fails the clinical trial in HCC therapy but success when combination with immune checkpoint inhibitor?
- Line 287-290, please specify which study used HCC tumor tissue and which used blood sample. The description may mislead readers.
Author Response
Reviewer #1:
This review paper discussed about the angiopoietin/Tie2 pathway in HCC. Generally it is well-written.
Comment 1: An illustrative figure portraits the complexity of this pathway and its associated pharmacological intervention (druggable node) will be helpful for readers.
Answer: We agree with the reviewer that addition of an illustrative representation of the involvement of the Ang/Tie2 signaling pathway in HCC angiogenesis will benefit comprehension of this matter. Therefore, we created a figure that illustrates the complexity of Ang-2-mediated vascular sprouting in HCC. As mentioned in the figure legend, targeting the components of this signaling pathway may be of therapeutic value for HCC patients.
Line 81: referred to figure in text
Line 82: figure
Line 83-97: figure legend
Comment 2: Summary subtitles can be placed ahead of each discussion main point to facilitate comprehension.
Answer: As recommended by the reviewer, we added subtitles in several chapters in order to enhance comprehensibility.
Chapter 4: Angiopoietin expression and signaling in HCC
- Line 124: 4.1 Angiopoietin-2 expression in HCC
- Line 169: 4.2 Angiopoietin-1 expression in HCC
Chapter 6: Angiopoietin-targeting therapeutic strategies in HCC
- Line 232: 6.1 Angiopoietin targeting in context of chemotherapy
- Line 257: 6.2 Direct angiopoietin-targeting therapeutic strategies
Chapter 7: Angiopoietins as diagnostic and prognostic biomarkers
- Line 317: 7.1 Angiopoietins as biomarker in non-systemic HCC treatment
- Line 352: 7.2 Angiopoietins as biomarker in sorafenib treatment
- Line 375: 7.3 Angiopoietins as biomarker in other systemic treatments
Comment 3: The rationale in line 54 and 55 may be misleading. A lot of clinical data showed that a single large HCC mass can be a well-differentiated tumor, instead of showing features of undifferentiation or dedifferentiation.
Answer: As large HCC tumors can indeed remain well-differentiated, we rephrased the sentence mentioned by the reviewer, in order to avoid misinterpretation.
Line 54-56: rephrased sentence
Comment 4: Can you discuss, with the molecular insights, why bevacizumab alone fails the clinical trial in HCC therapy but success when combination with immune checkpoint inhibitor?
Answer: As recommended by the reviewer, we have elaborated on the synergistic effect of the combination of immune checkpoint inhibition and anti-VEGF therapy. In addition, we discussed the potential value of Ang-2 inhibition in this context.
Line 69-78: added section on the combination of anti-angiogenic therapy and immune checkpoint inhibition.
Comment 5: Line 287-290, please specify which study used HCC tumor tissue and which used blood sample. The description may mislead readers.
Answer: As recommended by the reviewer, we have moved the references that relate to studies that used HCC tissue closer to the part they refer to, in order to improve comprehensibility.
Line 319-322: reference 22 and 63 are placed behind the section on HCC tissue, whereas reference 58 remains behind the section on blood samples.
Reviewer 2 Report
Review
In this review article, the authors overview the role of Ang/Tie2 pathway in HCC development and proposed the possibility as a target molecule and biomarker.
Comments
- The authors reviewed the molecular role of angiopoietin family molecules (Ang 1-4) in in vascularization. Please summarize the molecular mechanisms of them, especially Ang2, by making a figure to help understand the difference and connection between them.
- As authors described in line 67, the combination therapy of atezolizumab and bevacizumab showed the better prognosis than sorafenib and becomes the first line therapy of HCC. The significance of inhibiting vascularization under the treatment with immune checkpoint inhibitor should be discussed, and if possible, the effect of Ang2 inhibition in that situation may be mentioned.
Author Response
Reviewer #2:
In this review article, the authors overview the role of Ang/Tie2 pathway in HCC development and proposed the possibility as a target molecule and biomarker.
Comment 1: The authors reviewed the molecular role of angiopoietin family molecules (Ang 1-4) in in vascularization. Please summarize the molecular mechanisms of them, especially Ang2, by making a figure to help understand the difference and connection between them.
Answer: We agree with the reviewer that addition of an illustrative representation of the involvement of the Ang/Tie2 signaling pathway in HCC angiogenesis will benefit comprehension of this matter. Therefore, we created a figure that illustrates the complexity of Ang-2-mediated vascular sprouting in HCC. In this figure, we integrated the HCC-mediated quantitative balance shift between Ang-1 and Ang-2.
Line 81: referred to figure in text
Line 82: figure
Line 83-97: figure legend
Comment 2: As authors described in line 67, the combination therapy of atezolizumab and bevacizumab showed the better prognosis than sorafenib and becomes the first line therapy of HCC. The significance of inhibiting vascularization under the treatment with immune checkpoint inhibitor should be discussed, and if possible, the effect of Ang2 inhibition in that situation may be mentioned.
Answer: As recommended by the reviewer, we have elaborated on the synergistic effect of the combination of immune checkpoint inhibition and anti-VEGF therapy, and discussed the potential value of Ang-2 inhibition in this context.
Line 69-78: added section on the combination of anti-angiogenic therapy and immune checkpoint inhibition.
Reviewer 3 Report
The manuscript is a very focused and comprehensive review of the role of the angiopoietin and Tie2 signaling in liver cancer. Also, it is a well-written manuscript. The author summarized commonly used animal models to study angiopoietin signaling hepatocellular carcinoma (HCC) (Table 1). The author also summarized current treatment options for patients with HCC (Table 2). However, a visual schematic representation of the Ang-2/Tie2 signaling pathway will further increase the manuscripts' presentation, especially for visual learners, in a Figure form.
Author Response
Reviewer #3:
The manuscript is a very focused and comprehensive review of the role of the angiopoietin and Tie2 signaling in liver cancer. Also, it is a well-written manuscript. The author summarized commonly used animal models to study angiopoietin signaling hepatocellular carcinoma (HCC) (Table 1). The author also summarized current treatment options for patients with HCC (Table 2).
Comment 1: However, a visual schematic representation of the Ang-2/Tie2 signaling pathway will further increase the manuscripts' presentation, especially for visual learners, in a Figure form.
Answer: We agree with the reviewer that addition of an illustrative representation of the involvement of the Ang/Tie2 signaling pathway in HCC angiogenesis will benefit comprehension of this matter. Therefore, we created a figure that illustrates the complexity of Ang-2-mediated vascular sprouting in HCC.
Line 81: referred to figure in text
Line 82: figure
Line 83-97: figure legend